# Strategy and Challenges of Paraclinical Examinations in Adult-Onset Still’s Disease

**DOI:** 10.3390/jcm11082232

**Published:** 2022-04-16

**Authors:** Nicolas Poursac, Itsaso Odriozola, Marie-Elise Truchetet

**Affiliations:** 1Department of Rheumatology, FHU ACRONIM, University Hospital of Bordeaux, 33000 Bordeaux, France; nicolas.poursac@chu-bordeaux.fr (N.P.); itsaso.odriozola@chu-bordeaux.fr (I.O.); 2Immunology Laboratory, ImmunoConcept, UMR CNRS 5164, University Hospital of Bordeaux, 33000 Bordeaux, France

**Keywords:** adult-onset Still’s disease, biological examination, radiological examinations

## Abstract

Adult-onset Still’s disease is a complex autoinflammatory disease with a multifactorial etiology. Its presentation is less stereotypical than that of a monogenic autoinflammatory disease and is actually relatively common with few specific signs. To avoid under- or over-prescription of complementary examinations, it is useful to advance in a structured manner, taking into consideration the actual added value of each supplemental examination. In this review, we detail the different complementary tests used in adult Still’s disease. We consider them from three different angles: positive diagnostic approach, the differential diagnosis, and the screening for complications of the disease. After discussing the various tests at our disposal, we look at the classical diagnostic strategy in order to propose a structured algorithm that can be used in clinical practice. We conclude with the prospects of new complementary examinations, which could in the future modify the management of patients.

## 1. Introduction

Adult-onset Still’s disease (AOSD) is a rare inflammatory disease of unknown etiology [1]. Usually it affects young adults, and clinical manifestations are most commonly spiking fevers, arthritis, and evanescent rash [2]. Other manifestations that are frequently observed and useful to support the diagnosis are elevated liver enzymes, lymphadenopathy, hepatosplenomegaly, and serositis. However, as has long been debated, there is no single clinical, biological, histological, or radiological hallmark of the disease.

This lack of specificity often leads to a delayed or unrecognized diagnosis. It has been shown that a delay in diagnosis can influence future response to therapy. Kalyoncu et al., demonstrated that failure to obtain a rapid diagnosis leads to a chronic disease course in AOSD [3]. Similarly, optimal diagnosis and rapid prognosis assessment could reduce the main complications of the disease, such as macrophage activation syndrome (MAS), thrombotic thrombocytopenic purpura, respiratory distress syndrome, and diffuse alveolar hemorrhage [4]. Multi-visceral involvement of the disease and the different complications may strongly decrease life expectancy of AOSD patients. The absence of a key diagnostic element and the potential evolution towards a chronic form or visceral complications can lead to an excess of stressful paraclinical examinations. In the case of clinical presentations suggestive of hemopathy, investigations may include lymph node biopsies or splenectomy for diagnostic purposes. This can sometimes be avoided with a detailed knowledge of the clinical and paraclinical features of Still’s disease and a careful deductive organization of complementary examinations.

Classification criteria have been proposed and can help the clinician in the diagnostic process. Those of Yamaguchi published in 1992 are the reference [5]. The Fautrel criteria, which are easier to use, can also be applied [6]. A recent study has validated these two sets of criteria and has also proposed modified Yamaguchi criteria (which take into account ferritinemia and glycated ferritin), which allows for the reaching of a sensitivity and specificity of more than 90% [7].

Regardless of the availability of cutting-edge diagnostic tools, the medical history of the patient and an adequate physical examination remain at the forefront of the diagnostic strategy in clinical practice. In addition, some biological or radiological paraclinical tests are considered useful to complement the clinical examination. To complicate matters, a jungle of new paraclinical tests have arrived or are arriving in the equation. This highlights the urgent need to prioritize tests in line with an optimal strategy. AOSD remain a diagnosis of exclusion and the difficulty lies in the rational and adequate use of these exclusion tests.

In this review, we look at the individual value of different paraclinical tests and different strategic algorithms. Finally, we identify future challenges in the diagnosis of AOSD.

## 2. Paraclinical Examinations to Support the Diagnosis

Once interrogation and careful clinical examination have brought the physician on the track of Still’s disease, the long litany of paraclinical examinations begins. First to be prescribed are those which, if positive, will be able to support the diagnosis. The main tests and their use in the diagnosis of AOSD are summarized in Table 1.

### 2.1. Serological

The aim here is not to make an exhaustive list of all the biological tests that are used to make the diagnosis of Still’s disease.

We can start by pointing out that there is no biological result that formally proves the diagnosis. However, a biological inflammatory syndrome is necessary (even if it is insufficient) to confirm the diagnosis. The CRP is high and it is common to observe a ferritin level above 500. The release of ferritin—mainly by the histiocyte/macrophage system—is triggered by several cytokines, predominantly IL-1, IL-6, IL-18, and TNF. Still’s disease, whose pathophysiology is dominated by macrophagic activation and the role of IL1 and IL6, is an excellent model for ferritin secretion. Ferritin levels correlate with disease activity and resume on remission, yet the usefulness of ferritin as a diagnostic tool has been found to be limited (sensitivity 80%, specificity 40%). Since elevated ferritin levels are common in all inflammatory syndromes, this assay is only useful for diagnosis when it reveals an extremely high level. A threshold value of 1000 μg/L (five times the upper limits of normal (40–200 ng/mL)) has been proposed [8]. Before being secreted, part of the ferritin is glycosylated. In normal individuals and in the absence of any inflammatory condition, the level of glycosylated ferritin ranges between 50 and 80% of the total ferritin. Under inflammatory conditions and in Still’s disease, it frequently drops below 20%. This phenomenon is most likely due to the saturation of glycosylation mechanisms [9]. A fivefold rise in serum ferritin levels combined with a low level of glycosylated ferritin reaches a specificity of 93%. However, this anomaly persists for a long period of time, and therefore this parameter is not suitable for monitoring the follow-up [10].

If the infectious investigation is negative, a marked hyperleukocytosis, in particular more than 15,000/mm^3^ with neutrophilic polynucleosis greater than 10,000/mm^3^, can lend support to the diagnosis. However, this is not a universal condition, and in a retrospective study of 57 patients, it was present in about 70% of patients [2]. Seo et al., suggest the ratio of neutrophil count to lymphocyte count as diagnostic marker with a cut-off of value of 3 (i.e., a less drastic value than Yamaguchi’s criterion 4 and Fautrel’s criterion 5). According to Seo et al., it would be a very good way to distinguish an acute viral infection from Still’s syndrome, with an area under the curve of 0.967 [11]. On the contrary, a ratio of less than 3 should point to a diagnosis other than Still’s disease; in other words, a neutrophil percentage of less than 60% is not likely to be due to Still’s disease.

Individual susceptibility to Still’s disease has been investigated through the study of a potential link with certain HLA groups such as HLA-DR4 [12]. However, there is no evidence of any diagnostic or prognostic interest in performing this genotyping. Broader genomic studies are needed to look for mutations in other genes, particularly those in the inflammasome pathway. These will be carried out by aggregating patients into larger cohorts. Indeed, the relatively early onset of the disease even in adulthood suggests a larger genetic component than in other inflammatory rheumatic diseases.

Histology of dermatological elements is rarely used for the diagnosis of Still’s disease [13]. Cutaneous involvement occurs in about 80% of patients, including various clinical presentations. A constant histopathologic finding is the presence of interstitial dermal neutrophils aligned between the collagen bundles. Whilst this is not essential for diagnosis, and even unnecessary in typical forms, histological examination resulting in this pattern may provide an easily accessible clue for the definitive diagnosis of AOSD. It may also exclude other diagnoses such as drug eruptions or infectious diseases [14].

To summarize, biologically and diagnostically, it is of interest to confirm the patient’s inflammatory status (CRP, protein electrophoresis, blood count). An important element is a major increase in ferritin (above 4000 μg/L) associated with a glycosylated ferritin below 20%.

Cytokine assays, in particular IL-1 and IL-18, can be performed as part of the research but are not routinely offered. As described afterwards, demonstration of their diagnostic and prognostic value may turn them into the biological markers of the future for Still’s disease.

### 2.2. Radiological

Imaging of Still’s disease does not show specific features, but rather some radiological patterns. This can improve diagnosis strength, if used alongside clinical and biological signs mentioned above.

Rheumatologic involvement can affect all joints. Synovitis of large and medium joints is also frequently encountered in microcrystalline and autoimmune rheumatological disorders and is not discriminating. In contrast, axial involvement of the cervical spine; peri-capitate carpal destruction/fusion with metacarpophalangeal joints sparing [15,16]; trapezoid-metacarpal fusion; and, in young patients, distal interphanlangeal joints destruction [17] are much more specific, because only a few diseases are associated with this subtype of lesions [18]. Joints MRI patterns are not specific and show, as in rheumatoid arthritis, synovitis. Still’s synovitis would be characterized by intermediate-to-high signal intensity on T2-fat-saturated weighted and STIR images. MRI bone erosions and bone oedema could also be observed in around one third of the cases [19].

Pulmonary involvement is not common in Still’s disease, and CT scan lesions are not well documented due to the lack of large series of patients with Still’s disease and lung involvement. One study of 30 patients showed three main patterns: unclassified interstitial lung disease, organizing pneumonia, and non-specific interstitial pneumonia [20]. Another study on 30 patients described that the most common CT findings were peripheral consolidations and peribronchovascular consolidations. In that study, the main HRCT patterns were nonspecific interstitial pneumonia, organizing pneumonia, and unclassified interstitial lung disease [21]. The presence of lung disease could be a marker of severity and an emergent cause of mortality in AOSD.

Imaging data on damage to other organs are scarce and are often case reports. Periportally increased T2 intensity on the liver [22], myocarditis [23], or central nervous system inflammation [24] have been reported.

**Table 1 jcm-11-02232-t001:** The main tests used for the diagnosis of adult-onset Still’s disease (AOSD).

Marker	Description in AOSD	Usefulness for AOSD	References
CRP	>5 mg/L usually very high >50 mg/L	Not specific but essential for diagnosis	[4]
Polymorphonuclear neutrophils	>80% neutrophils among leukocytes	Cardinal criteria	[4]
Ferritin	>ULN often > 5 × ULN	High sensitivity if >ULN but high specificity (80%) only if >5 × ULN	[1,25,26]
Glycosylated ferritin	Low (<20%)	Sensitivity 79.5% Specificity 66.4%	[1,10]
IL-1β	Elevated but no standard and not different in sepsis	Not routinely used	[27,28,29]
IL-6	Elevated but no standard and not different in sepsis	Not routinely used	[28,29]
IL-18	No standard but levels >150 or 366 ng/L	Not routinely used Sensitivity 91.7% and specificity 99.1% when >366 ng/L	[30,31]
TNFα	Elevated but no standard and not different in sepsis	Not routinely used	[28,29]
Histology on skin biopsy	Broad histologic spectrum Interstitial dermal neutrophils aligned between the collagen bundles	Allow to exclude differential diagnosis in atypical forms	[13,32]
Joint X-ray	Peri-capitate carpal destruction/fusion with metacarpophalangeal joints sparing	Useful in advanced and articular forms of the disease/late-onset abnormality	[18]
Joint US	Active synovitis of large and medium joints	Useful in articular forms of the disease	[16]
CT scan/PET-CT scan	Lymphadenopathy, HSM/hypermetabolism in lymph node, spleen, and bone marrow	Not essential for diagnosis, useful for differential diagnosis	[33]

## 3. Paraclinical Tests to Exclude Other Diagnoses

In the diagnosis of adult-onset Still’s disease (AOSD), it is of the utmost importance to exclude differential diagnoses. The list of these diagnoses is quite extensive and includes first and foremost infectious diseases, but also neoplastic diseases and various inflammatory systemic diseases.

To avoid overloading the system with additional tests, it is useful and cost-effective to systematically carry out the most frequent serological tests (HIV, HBV, HCV, EBV, CMV), as well as blood cultures and a bacteriological examination of the urine. With regards to infectious diseases, other examinations should be carried out according to the patient’s context.

In the context of long-term fever with an altered general condition, lymphadenopathy and sometimes pruritus, the tumor hypothesis is a natural one, particularly that of hemopathies [34]. Lymphoma is one of the most difficult differential diagnoses to eliminate when faced with a picture suggestive of Still’s disease. Computed tomography (CT) of the chest/abdomen/pelvis should be performed in all cases. Histological samples and in particular an osteomedullary biopsy should only be taken if there is a suspicion of hematological malignancy. Similarly, a lymph node biopsy should only be performed in the case of asymmetric and/or indurated adenomegaly.

PET/CT is not systematically performed but may be discussed in some cases [33]. It should be noted, however, that even in cases of proven Still’s disease, PET/CT showed FDG uptake in bone marrow and lymph nodes in three out of four cases, and in the spleen in one out of two cases [35]. Sometimes the conclusion of the examination is in favor of a malignant origin because the spatial distribution of 18F-FDG uptake and its intensity can be similar to a malignant disease. We must therefore be very cautious with these conclusions, and for this reason, the place of PET/CT in the diagnostic algorithm is still difficult to determine. However, PET/CT could have other interests. The glucose metabolism of the liver, spleen, and bone marrow of AOSD patients have been correlated with inflammatory markers, suggesting that it could be applied to evaluate disease activity [36]. More interestingly, the prognosis could also be clarified by PET with a link between intense 18F-FDG uptake and monocyclic forms of the disease [35].

The question of Still’s disease often arises when an individual faced with a long-term fever. In a retrospective study, the authors analyzed 290 files and showed that when the cause of the fever was finally found (in almost 70% of cases), it was inflammatory in more than 35% of cases [34]. Within inflammatory diseases, some autoimmune diseases can mimic the main features of Still’s disease, although they usually cause lower CRP or ferritin levels [37]. A flare-up of lupus, for example, may present a significant lymphadenopathy, a very high CRP, and a skin rash. Such a profile may be difficult to differentiate from an inaugural flare-up of Still’s disease. In these highly inflammatory cases, progression to macrophagic activation syndrome can be possible [38]. An autoimmune work-up including at least anti-nuclear antibodies, anti-DNA, and ANCA (to rule out vasculitis, which may also have a highly inflammatory presentation) should therefore be performed [1]. The rest of the laboratory tests should be carried out according to the clinical context. A synovial fluid analysis should systematically be performed in case of arthritis.

## 4. Paraclinical Examinations to Detect Complications of AoSD

After the diagnostic difficulty, Still’s disease might appear to be a fairly mild form of inflammatory disease. However, a number of complications can affect the prognosis [39]. These complications, some of which may be life-threatening, must be detected as early as possible. This is why the paraclinical assessment must include a certain number of examinations to this end. They must either be carried out systematically or according to the clinico-biological profile of the patient.

Moderate liver involvement is frequently observed in Still’s disease. However, acute severe hepatitis is uncommon, but possible, and may be life-threatening. Liver function must therefore be monitored regularly. A recent study collected 21 cases of AOSD with severe hepatitis. They were mostly young adults and compared to patients without severe hepatitis they presented less arthritis, a macular rash, a sore throat, lymphadenopathy, or splenomegaly. Cytopenia was also more frequent in case of severe hepatitis [40]. The mechanism of hepatic necrosis in AOSD with severe hepatitis is unknown but can be the inaugural manifestation. However, the results of a liver biopsy are never specific to AOSD.

Secondary hemophagocytic lymphohistiocytosis (HLH) or macrophagic activation syndrome (MAS) occurs in around 10% of AOSD cases. As it is the most frequent and severe complication, it must be systematically probed for by at least a few simple serological tests (cytopenia, transaminases, triglycerides, fibrinogen) [41]. Its clinical features include a persistent high-grade fever, hepatosplenomegaly, lymphadenopathy, hemorrhagic manifestations, and a sepsis-like condition. In the case of suggestive clinical or serological abnormalities, a bone marrow sample must be taken and analyzed for hemophagocytes. The HScore, which was determined in a multicenter retrospective cohort study of 312 patients, is an indication of the likelihood of MAS being present. It relies on a set of nine variables: known underlying immunosuppression, high temperature, organomegaly, triglyceride, ferritin, serum aspartate transaminase, fibrinogen levels, cytopenia, and hemophagocytes features on bone marrow aspirate. MAS can be ruled out with an HScore of ≤90, whereas an HScore ≥ 250 has a diagnostic accuracy of >99% [42]. The serum-soluble interleukin-2 receptor (sIL-2r) level could be considered an important diagnostic test and disease marker in HLH. However, sIL-2r is rarely measured in clinical practice and has been excluded from recent diagnostic/classification criteria such as the HScore. In a retrospective study of 39 AOSD patients, 14 presented complications due to MAS. A higher prevalence of multiorgan involvement in AOSD patients with MAS has been shown, suggesting imaging-based differences and the importance of performing a CT scan [40].

Coagulation disorders with disseminated intravascular coagulation (DIC) or thrombotic microangiopathy are rare (less than 1–5% of patients), but should be monitored closely as they are particularly difficult to halt once they occur. In addition to the clinical signs of these events, close monitoring of platelets is imperative. In case of any suspicion of a coagulation disorder, it is useful to also monitor D-dimer, PT, and fibrinogen [43]. The diagnosis of acute DIC is made in the presence of thrombocytopenia, an increase in partial thromboplastin time (=TPP, TCK, TCA, TPP) and prothrombin time (TQ), an increase in D-dimer (and fibrin degradation products), and a decrease in plasma fibrinogen level.

Cardiopulmonary complications are particularly rare, but it seems that pulmonary hypertension is slightly underdiagnosed. In this context, a systematic echocardiography could be discussed. This is not yet standard practice, and thus far it is only performed in case of suggestive symptoms or to rule out differential diagnoses such as infectious endocarditis [44].

In chronic forms high levels of serum amyloid A, especially if it remains high while the CRP is low, can predict the development of systemic amyloidosis [4]. However, the development of amyloidosis is extremely rare in adult still disease and should not be routinely investigated [45].

## 5. Proposal for Stratification of These Tests

After this review of the literature, we propose an algorithm for carrying out complementary examinations when Still’s disease is suspected (Figure 1). The initial clinical signs that allow entry into the algorithm are a persistent fever associated with skin rash and arthritis. When faced with an atypical presentation, the diagnosis must be evoked after having eliminated the other diagnoses even more exhaustively. Small signs such as odynophagia, joint involvement, skin rash, and fever rhythm can be valuable indicators. Finally, the interleukin-18 test has shown promising results and may one day be used in clinical practice [31].

## 6. Future and Perspectives of Paraclinical Examinations for AOSD

In this review, we outlined what is currently being done to optimize the diagnosis of Still’s disease. However, there are still many unknowns and uncertainties regarding the diagnosis and prognosis of this disease. New avenues for further investigation are being explored, and some of these may be the routine of the future.

The study of the pathophysiology of Still’s disease has revealed that certain cytokines are at the core of the mechanism underlying the onset of the disease. Some teams have attempted to monitor these cytokines. In that field, IL-18 is a very interesting candidate. Some teams have shown, for example, that plasma IL-18 associated with ferritin levels, or that fibroblast growth factor 2 could be used to differentiate sepsis from AOSD [46,47]. More recently, it has been shown that IL-18 could also be a differentiation biomarker between AOSD and COVID-19 [30]. Free IL-18 could be an efficient marker for diagnosis and follow-up in AOSD patients and may be a useful predictive marker of remission, notably in the inactive stage [48]. Interleukin-37 would also be of interest in this context [49]. Interestingly, patients may have different cytokine profiles depending on their clinical presentation, but also on their prognosis. Patients with high levels of IL-18 would require higher doses of corticosteroids to go into remission and would present forms associated with pleuritis and significant hyperferritinemia [50]. These promising observations need to be extended to larger cohorts as they could help the clinician in the diagnosis and initial therapeutic decision.

Other biomarkers are regularly studied as physiopathological knowledge advances. Some of them give promising results (heparin-binding protein, serum S100A8/A9 that correlate with leukocyte count, ESR, CRP, ferritin, and systemic disease score), while others remain controversial [51]. For instance, serum calprotectin is considered to be quite interesting at differentiating AOSD patients from patients with other rheumatic diseases and healthy controls [52].

A real paradigm shift for therapeutic decision could be the application of systems biology to AOSD. A recent preliminary study with systems biology-based modeling highlighted the preferred use of biologics as an immunomodulatory treatment strategy for Still’s disease and reinforced the importance of IL-1 blockade for innate immunity regulation [53].

Overall, these markers have good diagnostic sensitivity and are indeed elevated in Still’s disease with varying degrees of prognostic value. The challenge lies in their specificity. None of them is really increased exclusively in Still’s disease. Probably the most promising strategy will be to define multiparametric profiles that minimize the risk of error.

## Figures and Tables

**Figure 1 jcm-11-02232-f001:**
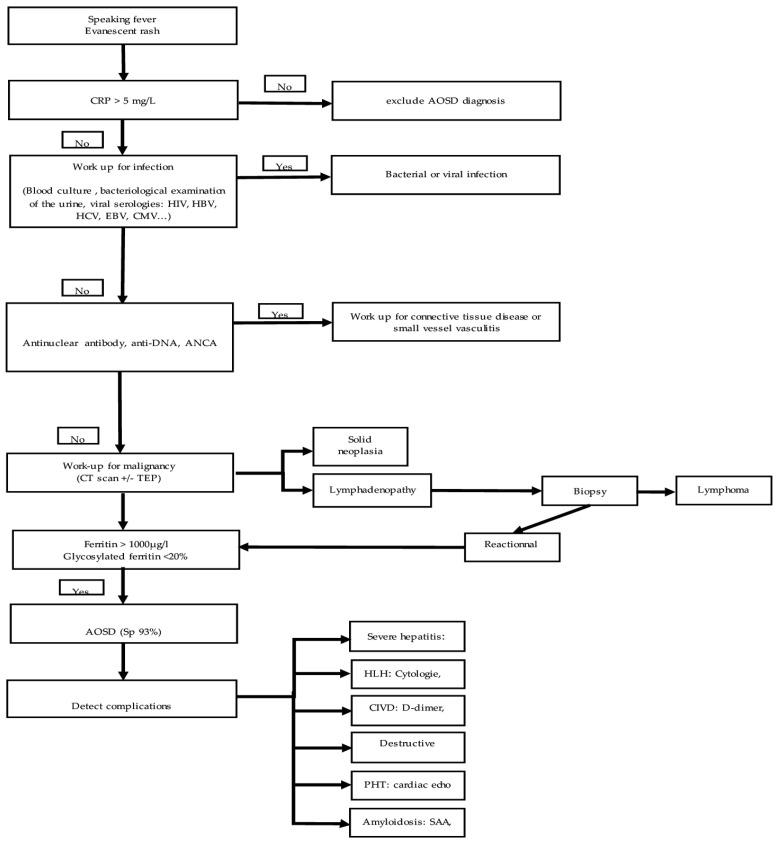
Proposal for stratification of complementary examinations in the presence of a clinic suggestive of Still’s disease. Legend: AOSD, adult-onset Still’s disease; HIV, human immunodeficiency virus; HBV, hepatitis B virus; HCV hepatitis C virus; EBV, Epstein–Barr virus; CMV, cytomegalovirus; ANCA, anti-neutrophil cytoplasmic autoantibody; TAP-Scan, thoraco-abdominopelvic scan; PET-Scan, positron emission tomography scan; ADP, adenopathy; HSMG, hepatosplenomegaly; PT, prothrombin time; HLH hemophagocytic lymphohistiocytosis; Tg, triglycerides; Fg, fibrinogen; DIVC, disseminated intravascular coagulation; TMA, thrombotic microangiopathy; Rx, radiography; US, ultrasonography; MRI, magnetic resonance imaging; PHT, pulmonary hypertension; SAA, serum albumin A.

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
