# Peer review of "Strategy and Challenges of Paraclinical Examinations in Adult-Onset Still’s Disease"

_jcm, 2022, doi:10.3390/jcm11082232_

Round 1
Reviewer 1 Report
- I do not understand why you classified essential serologic or imaging studies for the diagnosis of AOSD as paraclinical exams. As I understand, we usually do not classify common serologic or imaging studies like CT scan as paraclinical exams.
- Line 17: I do not agree that the structured algorithm you suggested is anything new. Rather, it seems close to the standard protocol that most references recommended.
- line 40: I recommend to comment several paraclinical examinations as examples.
- Line 62: Most clinicians are not familiar with the term, ‘Biological’, when referring to tests listed in this paragraph.
- Line 75 & Figure 1: Thresholds for serum ferritin level are different. In addition, how would you diagnosis patients with AOSD whose serum ferritin levels are below 4,000?
- Line 149: what is VIH ?
- Line 153: As I understand, adenomegaly is not commonly used medical term in the clinical field.
- Line 156: thoraco-abdomino-pelvic scan does not seem as commonly used medical term in the clinical field.
- Line 157-159: I recommend you to list appropriate references accordingly. In what situations do you suspect the possibility of hematological malignancies?
- Line 164-166: I recommend you to list appropriate references accordingly.
- Line 174-178: This paragraph needs more detailed explanations with appropriate references.
- Line 177: I recommend using the term ‘synovial fluid analysis’ rather than ‘joint puncture’.
- Line 203-204: Could you check the accuracy of sentences here?
- Figure 1: Biological inflammatory syndrome might be unfamiliar medical term for most rheumatologists. I would rather use more detailed information here.
- Figure 1: How about you to put exams for infections altogether?
- Figure 1: How about you to put exams for autoimmune diseases altogether?
- Figure 1: Positive ANA or ANCA results do not always indicated the possibility for specific autoimmune diseases. They are not specific tests. I would rather comment ‘work up for other systemic autoimmune diseases’.
- figure 1: TAP scan and ADP do not seem as commonly used medical term in the clinical field.
In addition, I would rather comment this part as ‘work up for malignancy’.
CT scan also could be included in exams for excluding the possibility of infections.
In case of hepatosplenomegaly, where do you perform biopsy?
In what cases do you perform lymph node biopsy?
- Figure 1: there must be many patients with AOSD whose serum ferritin levels are below 4,000. How would you deal with these patients?
Reviewer 2 Report
The topic of the review is an important one; physicians and rheumatologists are increasingly asked to review patients with features which could be consistent with AOSD. However, the review is not entirely successful in its aim.
Throughout the text there are mentions of abnormalities in various measures, but not a description of the abnormality, or how this could be used to distinguish from other causes. A summary table may be helpful. Although statements are provided with references, in most cases I felt I would need to go and read each reference to work out how to interpret the tests recommended.
More discussion on recognition of secondary HLH would be helpful. What is the roles of soluble IL-2 receptor?
Page 5: recommendation to monitor d dimer and fibrinogen - these may be elevated in many inflammatory conditions and may not always indicate thrombosis. What changes would be concerning?
Figure 1 is helpful but practically many patients who are referred with suspected Still's do not have classical features e.g. patients on the intensive care unit with ongoing inflammatory response despite treatment for infection How could these patients be fitted into the algorithm?
References are all numbered twice.
In general, the paper is extremely difficult to read and would benefit from edits to style to improve readability.
For example, the sentence in para 2 ‘Unfortunately, this initial phase would be crucial for the quality of subsequent management, as the response rate to treatment may depend on it.’ Could be replaced by ‘This can lead to a considerable delay in diagnosis which can influence future response to therapy.’
There are a number of grammatical errors throughout the text e.g. Joint’s rather than joints, lack of capitalisation at the start of some sentences (page 4 line 23, page 7 line 257)
Abbreviations should be defined on first use, or not used at all if only required once, and are different between the text and figure. Is VIH human immunodeficiency virus, i.e. HIV? Does TAP scan refer to a CT scan - this is unclear in fig 1 description
Round 2
Reviewer 1 Report
Thank you for the improvement in your manuscript.
- Table 1 : cyclosylated ferritin -> glycosylated ferritin
- Figure 1: I can only see the previous version of Figure 1. Did you modify the Figure 1 accordingly?
Author Response
April 2022
Ms. Sibyl Dai
Assistant Editor
Journal of Clinical Medicine
Re: Manuscript ID jcm-1653882 (by Poursac N. et al. “Strategy and challenges of paraclinical examinations in Adult-onset Still’s diseases”).
Dear Ms Dai, dear reviewer,
Please find below our secondpoint-by-point response concerning the manuscript entitled " Strategy and challenges of paraclinical examinations in Adult-onset Still’s diseases " (jcm-1653882).
We believe that we have completely and carefully answered the questions asked by reviewer 1.
We hope that you will find the final clarifications in this point-by-point answer.
Best regards,
- Table 1 : cyclosylated ferritin -> glycosylated ferritin
We thank the reviewer for this comment and we made the change in table 1.
- Figure 1: I can only see the previous version of Figure 1. Did you modify the Figure 1 accordingly?
We have made all the changes in Figure 1, but we were also asked to provide a better quality file. It is therefore uploaded outside the manuscript as a TIFF file.
Thank you again for your comments.